# Prepubertal Use of Long-Term GnRH Agonists in Dogs: Current Knowledge and Recommendations

**DOI:** 10.3390/ani12172267

**Published:** 2022-09-01

**Authors:** Sabine Schäfer-Somi, Duygu Kaya, Selim Aslan

**Affiliations:** 1Platform for Artificial Insemination and Embryo Transfer, Vetmeduni Vienna, 1210 Vienna, Austria; 2Department of Obstetrics and Gynecology, Faculty of Veterinary Medicine, University of Kafkas, 36100 Kars, Turkey; 3Department of Obstetrics and Gynecology, Veterinary Faculty, Near East University, 99138 Nicosia, Cyprus

**Keywords:** dog, GnRH agonists, deslorelin, puberty

## Abstract

**Simple Summary:**

Delay of puberty is sometimes desired by pet owners and can prevent early occurrence of estrus and unwanted pregnancies. Long-term delay of puberty is possible with GnRH agonists such as deslorelin that can be applied as subcutaneous implants. Many studies have been performed to investigate the best age for implantation, and whether undesired side-effects on body development, height at whither, and behavior may occur. At present, most important appears the age at implantation in that one 4.7 mg deslorelin implant should be inserted s.c. between the 3rd and 4th month of life to prevent a flare-up or induction of estrus. Body development as well as fertility after occurrence of first heat after the end of efficacy were normal. However, long-term effects on joint health, tumor development, immune system, and behavior deserve further investigation.

**Abstract:**

The search for an alternative approach of estrus control (induction or suppression) in dogs is an important issue and the use of slow GnRH agonist-releasing implants has been the subject of frequent research in recent years. Studies to date demonstrate that the short- and long-term effects of deslorelin implants applicated at different time points of the prepubertal period are similar to those of adult dogs; however, there are important differences. The age of the prepubertal bitch and the dosage appear to be the main determinants of the response to deslorelin, as well as the individual metabolism of the bitch. Recent studies reported that the deslorelin-mediated long-term delay of puberty does not have negative carry-over effects on subsequent ovarian functionality, serum steroid hormone concentrations, uterine health, and fertility; however, more molecular studies are needed to determine the effects of application time of GnRH agonists on hormone concentrations and peripheral receptor expression. Furthermore, the long-term effects of delay of puberty with deslorelin on joint health, tumor development, the immune system, and social behavior deserve further investigations.

## 1. Introduction

Gonadotropin-releasing hormone (GnRH) is a decapeptide hypothalamic hormone affecting GnRH receptors in the pituitary. GnRH stimulates production and secretion of both luteinizing hormone (LH) and follicle-stimulating hormone (FSH), affecting the gonads regulating steroid production, spermatogenesis, ovarian follicular development, and ovulation [1,2]. GnRH agonists are produced with amino acid substitutions of the natural GnRH molecule to create a longer duration of action and a long half-life. Prolonged application of GnRH and GnRH agonists leads to desensitization and downregulation of GnRH-receptors in the pituitary gland, finally suppressing gonadotropin secretion and inhibiting the pituitary–gonadal axis, which was also proven in dogs [3,4]. The development of slow-release GnRH agonists has a long history, but meanwhile subcutaneous implants are available and are supposed to exert a long-term effect over several months, for fertility control of male and female dogs. These products can mostly be used for both induction and suppression of estrus in female dogs [2,4,5,6,7].

One of these slow-release GnRH agonists is deslorelin; the 4.7 mg implant was initially licensed for contraception in male dogs and the 9.4 mg implant for the same purpose in male dogs and ferrets. Meanwhile, deslorelin was successfully used off-label to postpone puberty without affecting ovarian function and body development in male and female dogs [8,9,10,11,12,13]. In prepubertal bitches of medium body size, both the 4.7 mg and the 9.4 mg deslorelin implant were found to delay puberty until >2 years of age [11]. Later on, the local impact of deslorelin on ovarian suppression and resumption of ovarian activity was highlighted, and no negative effect on ovarian functionality found after recurrence of cyclicity [12]. In the uterus, no abnormalities were found during suppression and after the first heat, in comparison to non-treated controls [13]. Even though the long-term effects of delay of puberty with deslorelin on joint health, tumor development, the immune system, and social behavior deserve more investigations, the results so far are promising. Especially the fact that the increase in LH and FSH concentrations observed after gonadectomy is prevented by using a subcutaneous GnRH agonist implant is interesting. Recent findings emphasize a probable negative effect of the increased LH concentrations on the health of gonadectomized dogs [14].

This review mainly focuses on the current knowledge on the prepubertal use of deslorelin as an alternative to early surgical sterilization in dogs. Old and new findings in both male and female dogs on the use of deslorelin-releasing implants at different time points of the prepubertal period inclusive those on the molecular base are reviewed.

## 2. Review

### 2.1. Puberty

Puberty is defined as the process of physical changes by which an animal matures into an adult capable of sexual reproduction. The age at puberty is highly variable, even within breeds and independent on body weight [15]. The onset of first estrus indicates the beginning of puberty in females. In males, it is the age at which the first full ejaculate can be collected [15]. However, as reviewed by Gobello [16], puberty probably starts earlier, in females, marked by follicular growth, a change in vaginal cytology, and the increase in steroid hormones or precursors such as dihydroepiandrosterone DHEA [10,17]. In males, it is sometimes marked by beginning of mounting or single spermatozoa in the urine or ejaculate [16].

To be sure that indeed prepubertal dogs are investigated, studies including prepubertal animals should follow strict guidelines. A thorough history should be taken concerning the behavior, then a gynecological examination including vaginoscopy and vaginal cytology should be done. Using a tiny, moisturized vaginoscope, it is possible to evaluate whether the mucus membranes are edematous or not and by taking a swab it is possible to do a cytological examination; an increase in the superficial cells will indicate the beginning of cytological estrus [18]. This is furthermore useful to exclude a juvenile vaginitis. This inflammatory condition usually resolves spontaneously after the first estrus; however, it will persist in most cases when the bitch is spayed [15]. In one study [18], six prepubertal bitches with juvenile vaginitis that had received a deslorelin implant at the age of 3 months showed spontaneous remission after the first treatment, which was supposed to be caused by the initial increase in E2. Nevertheless, in future studies, these bitches should be excluded since not all bitches that received a GnRH agonist before puberty showed an increase in E2.

In male dogs, after taking a thorough history, an andrological examination should be done. Finally, in both male and females, measurement of sexual steroids is necessary to determine whether the animal is in fact prepubertal. In the present review, while comparing the results, these parameters as well as the age at treatment were considered.

Measurement of AMH can be helpful; however, in one study, 50% of female dogs aged 3–6 months (n = 14) had a serum-AMH concentration higher than the reported cut-off value, the other 50% were below this value. At the age of 6 months, 93.9% had a serum-AMH concentration higher than the reported cut-off value [19]. Therefore, the measurement of AMH seems to be more reliable in bitches older than 6 months as a test for intactness than as a proof of prepubertal ovaries. The latter should be confirmed by examination of more prepubertal bitches.

### 2.2. Postponement of Puberty in Female Dogs

Studies investigating the postponement of puberty are difficult to compare because of high variability considered in the different study designs. Readers furthermore have to take care whether time to puberty (after implant insertion) or the age at puberty was recognized, since both terms are used. The definition of flare-up is inconsistent; in some studies, it is just an increase in cornification of superficial cells and an increase in serum-estradiol (E2) concentrations [11,18,20]; in others, it is vulvar swelling and/or vaginal discharge, sometimes combined with an increase in superficial cell index (SCI) and/or an increase in serum-progesterone concentrations [8,11,20].

Furthermore, the GnRH agonists vary. Concannon [21] described the effect of subcutaneous osmotic pumps constantly delivering a GnRH agonist related to nafarelin over one year; others applied 100 µg of an injectable GnRH agonist SID over 23 months. Furthermore 18.5 mg azagly-nafarelin [7], as well as 4.7 mg [11,18,22], 9.4 mg [8,11,22], and 18.8 mg deslorelin [23] as implants were used. Some studies were performed without a control group [10,22].

### 2.3. Age at Implant Insertion—Prepubertal or Not?

Comparison of the age at the beginning of the study reveals that the youngest individuals received the implant at the age of one day [23], and some at the age of 5.3 months [22]. However, the age at the beginning of puberty is an important factor since in beagle dogs, the increase in FSH and LH as well as DHEA occurred as early as 4 months of age [24]. Insertion of a deslorelin implant in bitches older than 4 months, as was done in some studies [8,22], may therefore be too late and the results confounded. Unfortunately, in some studies, the examination of the bitches before and during the treatment is insufficiently described or performed. Table 1 shows the available literature about postponement of puberty in bitches and reveals the different methods to confirm puberty.

In some studies, the bitches had been examined clinically including vaginal cytology and hormones such as P4, E2, DHEA, and other steroid hormones were measured [10,11,18,20]. In one study, radiographic examinations were performed monthly to determine epiphyseal closure [11]. This seems more favorable than mere observation of vulvar swelling and vaginal bleeding [7] since the first signs of puberty could be overlooked.

### 2.4. Flare-Up or Not?

A flare-up with visible effects of estrogens was described in four out of seven studies in female dogs, indicating that in some dogs, FSH secretion and follicular development had already started. Sometimes a vulvar swelling was the only sign indicating a flare-up [9], in some cases accompanied by an increase in superficial cell index (SCI) or an increased percentage of cornified cells [8,11,18,20]. Bloody vaginal discharge indicating estrus bleeding only occurred, when puppies aged ≥7 month received a subcutaneous deslorelin implant [4]. In three out of seven studies, the measured increase in estrogens was not accompanied by other flare-up symptoms [20]. A temporary increase in progesterone >2 ng/mL without ovulations was noticed in single bitches and independent of the age at implant insertion [4,11,20]. In the study of Faya et al. [23], none of the newborn puppies that had received subcutaneous deslorelin implants at the age of one day showed flare-up symptoms.

### 2.5. Duration of Postponement of Puberty

When 100 µg of an injectable GnRH agonist were given as SID over 23 months, puberty of nine clearly prepubertal bitches was delayed until 4 to 12 months after cessation of the treatment [10]. This study highlighted the efficacy of GnRH agonists; however, later on, the long-acting subcutaneous GnRH agonist implants were introduced and proved to be highly effective in many studies.

In one of these studies, bitches were randomly allocated to treatment or control group at the age of 4.88 ± 0.32 months [7]. The experimental bitches received 18.5 mg azagly-nafarelin subcutaneous implants. No bitch showed signs of a flare-up, as indicated by vulvar swelling and vaginal bleeding, and cyclicity was suppressed over 24 months. This is the only available study where the implant was removed and after one year. However, to make sure that a flare-up during treatment did not occur, regular vaginal cytology and estrogen measurements would have been advantageous. This is mandatory for the long-term effects of the GnRH agonists to be comparable among studies.

In another study [8], the puppies were thoroughly examined before the application of the implants (9.4 mg deslorelin). Pups were randomly allocated to three groups and either received the implant at the age of 4 months or at the age of 7 months; a third group received a placebo at the age of 4 months. Vulvar swelling, sanguineous vaginal discharge, increasing percentage of superficial cells, and/or serum progesterone >2 ng/mL were noted as signs of puberty. This early study clearly showed that when the implant was inserted at the age of 4 months and in prepubertal bitches, none of these signs occurred within the observation period; unfortunately, the latter was only 9 months. A longer observation period would be desirable to enable evaluation of the medium- and long-term effects of puberty delay.

Fontaine et al. [22] were the first to use a 9.4 mg deslorelin implant for postponement of puberty in bitches. The bitches were aged on average 4.2 months; 11 received the 9.4 mg implant, 16 other bitches the 4.7 mg implant. Unfortunately, a body weight range between 3 and 40 kg is given, but not the average body weight and it is not clear which dogs received which implant. However, a more homogeneous group would simplify the subjective evaluation of the effects. The effect in very large breeds may be questioned due to the naturally late-occurring puberty in some individuals with high body weight. In this study, the owners were asked to call the clinic when they observed signs of estrus. Examination of the bitches in estrus was done at the end of the estrus, side effects noted, and progesterone measured. In the group of bitches that received the 9.4 mg implants, none showed signs of estrus within the observation period of 8–15 months; however, a flare-up can be easily overseen by owners. Earlier examinations using vaginal cytology and estrogen measurement would have been advantageous to recognize the beginning of follicular development and occurrence of puberty.

Later on, Kaya et al. [11], using 4.7 mg implants in 4 bitches aged 4.3 months and 9.4 mg implants in 5 bitches aged 4.2 months, observed occurrence of puberty at the age of 18 to 25 months. The deslorelin concentration had no effect on the duration of suppression; however, this might be biased since the low patient number is a weakness of the study. In the four control bitches that received 0.9% NaCl, puberty occurred after 15 months on average.

In the study by Faya et al. [23], cross-bred dogs with an average body weight of 13 kg were mated, and the newborn male and female puppies were randomly assigned to experimental and control group at the age of one day. The treatment group received two 9.4 mg implants subcutaneously in the left and right shoulder blade region. They were examined once weekly for body development (height at wither, body weight, scrotum circumference) and signs of estrus; and blood was taken for estradiol (E2) and testosterone (T) measurement. In addition, the owners looked for sexual behavior. The controls received a subcutaneous placebo implant. The age at puberty was almost double in the treatment group (72 weeks) in comparison to the controls (39.3 weeks) and was comparable to the effect of treatment in 4–5 months old puppies [11,18,22]. One advantage of the study is that these puppies clearly were prepubertal at the age of implant insertion; however, the necessity of such an early treatment may be questioned. Furthermore, evaluation of long-term effects on the sexual organs will be necessary, since side effects such as perivulvar dermatitis and ascending infections of the urogenital tract frequently occur after prepubertal castration [25,26,27]; observations during the first years of life will clarify whether these side effects can be prevented by using a long-term GnRH agonist.

Finally, in the cited studies, the time to puberty ranged between 9 and >25 months, the age at puberty between 18 and 31 months, and in some individuals, the suppression lasted even longer. In some cases and some breeds, the duration of efficacy will not exceed the natural time to puberty, since there is a broad range within breeds. Furthermore, the duration of efficacy is difficult to foresee and may be dependent on other factors such as site of application, resorption, and individual metabolism, as in adult dogs. It is still unclear, whether in bitches, a dose-efficacy relationship exists, since the duration of efficacy was similar with 4.7 and 9.4 mg deslorelin implants [8,11,18,20,22]. Nevertheless, postponement of puberty for several months to more than two years is possible.

### 2.6. Body Development

In most studies, experimental animals and controls had equal wither height at puberty and normal body weight [7,10,11,20,22,23]. This is different from earlier reports about prepubertal gonadectomy; the latter caused delayed closure of growth plates in cats and dogs, and an increase in the length of the radius [27,28]. Even though this was expected to be similar after prepubertal use of deslorelin implants, it was not observed in any study. A local effect of sexual steroids on the growth plate during the flare-up period cannot be excluded when GnRH agonists are used for postponement of puberty. This may vary between individuals as the flare-up symptoms are individual, but as a matter of fact, the average height at withers was not increased after the treatment. Probably, measurement of height at withers should always be accompanied by X-rays of the epiphyseal cleft, which clearly indicates the end of growth. Nevertheless, despite this supposed local effect of sexual steroids, the occurrence of estrus inclusive vaginal bleeding and ovulation were clearly prevented until the age of 18–31 months.

In the study by Kaya et al. [11], radiographic examinations were performed monthly until epiphyseal closure was observed. Authors noticed normal body and external genitalia development; however, epiphyseal closure was clearly delayed. Similar to the supposed local effect on the growth plates, a local effect of sexual steroids on the vulva cannot be excluded. The sensitivity of estrogen receptors increases during puberty; receptor density and individual sensitivity might play a role as well. Additional measurement of DHEA and/or more sensitive hormone assays might be helpful, and a larger patient number will be required to reveal whether the normal development of the vulva is a rule. However, as a matter of fact, the vulva remained juvenile, when three 4.7 mg deslorelin implants were inserted subcutaneously at the age of 4.5, 9, and 13.5 months [18]. This points towards a more effective suppression of sexual steroid hormones; the age at insertion and the time to puberty was similar between the two studies [11,18], thus one implant would be preferable, provided the delayed epiphyseal closure does not cause orthopedic problems later on.

Interestingly, even when neonatal bitches received 18.8 mg deslorelin at the age of one day, height at withers was similar between experimental animals and controls at the end of the observation period of >100 weeks [23]. This is difficult to explain; however, since the average time of suppression was 72 weeks in the experimental group, this could well indicate that in some individuals, there was no effect at all. This age at first puberty is described as within the normal range in dogs of this body weight class [15]. Unfortunately, development of the vulva and vaginal canal are not described.

### 2.7. Effects on the Uterus, Ovaries, and Fertility

Some studies showed that after postponement of puberty with both 4.7 and 9.4 mg implants bitches showed normal ovarian function, uterine health, and fertility [10,11,12,22,23]. The local impact of deslorelin on ovarian suppression and resumption of ovarian activity was examined after the end of suppression of cyclicity [12]; Kaya et al. [12] investigated enzymes and hormones involved in the corpus luteum (CL) establishment and function. Authors observed no long-term effect on luteal function after the first estrus. Expression of enzymes involved in steroidogenic pathways in the ovaries such as steroidogenic acute regulatory protein (*STAR*) was similar in both groups, even though expression of another enzyme involved in these pathways, namely 3β-hydroxysteroid dehydrogenase (3ßHSD), was significantly increased after deslorelin treatment. However, the progesterone concentration during metestrus following the first estrus after suppression was equal in both groups [12]. Similarly, expression of enzymes involved in prostaglandin synthesis pathway like cyclooxygenase 2 (*COX-2* and *PTGS)*, did not differ significantly between groups; the prostaglandins are significantly involved in the establishment of CL function in dogs [29]. In the study by Kaya et al. [12], kisspeptin receptor (*kiss1-R)* and GnRH-R were not significantly changed in the first cycle following suppression; similar to prolactin receptor (PRLR), which is upregulated during early- and mid-luteal phase to enable the luteotropic action of prolactin. Progesterone receptor *(PGR),* estrogen receptor alpha *(ERα),* and estrogen receptor beta *(Erß)* were not different between groups either. Altogether, delay of puberty at the age of 4 months did not have a negative effect on ovarian and corpus luteum function in the first estrus following suppression with deslorelin [12]. In the uterus, no abnormalities were found during suppression or after the first heat in comparison to non-treated controls [13]. In a follow-up study, the same dogs as in the study of Kaya et al. [12] were used; the uterine tissue was examined for expression of GnRH receptor (R), Kisspeptin (KP)10, Kisspeptin receptor (GPR54), androgen receptor (AR), estrogen receptor (ER) α, β, and progesterone receptor (PR) by means of immunohistochemistry and PCR. The treatment obviously had no significant impact on the expression of these receptors important for sexual hormone effectiveness in cycling dogs. No macroscopical and microscopical changes were seen in the uterine tissue, and no difference was found between deslorelin-treated bitches, operated after their first estrus, and the controls. Delay of puberty furthermore did not influence fertility after resumption of cyclicity [10,23]; even two of the puppies that received 18.8 mg deslorelin during the first 24 h of life became pregnant after mating during the first estrus following suppression [23]. Similarly in adult female dogs, deslorelin did not influence fertility after resumption of cyclicity after estrus suppression, provided the following cycle was normal [30]. Even though so far only 17 dogs treated before puberty were evaluated for consecutive fertility [10,23,30], a normal function of the pituitary–gonadal axis and uterine health can be expected [10,13].

### 2.8. Miscellaneous Effects

No side effects were observed when the implant was inserted between the 4th and 5th month of life, except the flare-up symptoms in some cases that stayed without bloody vaginal discharge.

The consecutive decrease in the blood concentration of FSH and LH is one important difference when the effect of prolonged application of GnRH agonists is compared to gonadectomy. After gonadectomy, serum-concentrations of FSH and LH stay elevated for years if not decades. The probably negative, local effects of these increased concentrations on different organs and cancer development are currently under investigation [14,31]. It is of interest whether the long-term suppression of cyclicity with slow-release GnRH agonists influences the incidence of some of the mentioned diseases, probably related to increased LH receptor expression after gonadectomy [14]. This would be a desirable effect in male and female dogs; however, further studies and observation of treated dogs over years are necessary to provide evidence.

### 2.9. Effects on Behavior in Female Dogs

In both female and male adult dogs, behavioral changes after chemical castration with a GnRH agonist, mostly occur as a result of transient sexual stimulation (flare-up) following application of a GnRH analogue such as deslorelin [32,33,34]; these changes are therefore mostly sexual. Although this period varies individually, it is mostly observed in the first two weeks and then disappears with suppressed gonadotropin release. In order to prevent induction of estrus, different treatments such as changing the implant insertion time, using exogenous progestins, acyline, anastrozole, clomiphene acetate. or osaterone acetate have been used and controversial, non-satisfying results were obtained [35,36,37]. Similar to male dogs, it has been reported that GnRH agonist implant applications in the early prepubertal period (mean age 4–4.5 months) do not cause flare-up symptoms in females [7,8,11,20]. It was observed that deslorelin applicated after 7 months of age, caused estrus with clinical behavioral changes in all (6/6) female dogs [8]. In our recent clinical study, performed in the late prepubertal period (7 to 8 months of age), we detected estrus symptoms in only 6/16 dogs (37.5%) who received a 4.7 mg deslorelin implant. Aggression was observed during proestrus in one of these dogs, while a significant increase in food intake was observed in four dogs. Behavioral changes were not detected in the other 10/16 dogs who did not show clinical signs of estrus after implant insertion. All behavioral changes were therefore caused by the flare-up. Physical and behavioral changes associated with false pregnancy were not observed in any of the late prepubertal dogs (Kaya et al., unpublished data).

A recent cohort study showed that female Labradors and Golden Retrievers, neutered before puberty, were more likely to have aggression factor scores that increased between the 1st and 3rd year of life; however, only a small number of bitches displayed only mild aggressive behaviors. There was no impact of pre- or post-pubertal neutering on other behavioral factors [38].

According to recent empirical data, experiences, and studies, the effect of deslorelin on behavior of prepubertally treated bitches is mild; as an advantage, the effect in case of exceptional changes in behavior is reversible.

### 2.10. To Summarize the Current Knowledge on Deslorelin in Female Dogs

The age at implant insertion should be between 3 and 4 months for small and medium-sized dogs.

Puberty should be excluded by clinical examination inclusive of vaginal inspection and vaginal cytology, estrogen/DHEA measurement, and lack of sexual behavior.

The concentration of deslorelin seems not to be related to the duration of efficacy.

It is not necessary to remove the implant.

Body development was normal in all studies when only one 4.7 or 9.4 mg implant was used.

Epiphyseal closure was delayed; the impact on joint health should be further investigated.

### 2.11. Postponement of Puberty in Male Dogs

Few studies have been performed with male dogs and with low patient numbers. Table 2 provides an overview over the results so far.

As in female dogs, studies are difficult to compare and it has to be differentiated whether time to puberty (after implant insertion) or the age at puberty was recognized. The age at the beginning of the study varies between “within the first 24 h of life” [23] and >5 months [10]. Furthermore, the GnRH agonists vary; in one study, 100 µg of an injectable GnRH agonist was given SID over 23 days [10], in the other, both 4.7 and 9.4 mg implants were used [9], and in the last study, two 9.4 mg implants were inserted at the same time [23]. All studies are cohort studies including a control group.

### 2.12. Age at Implant Insertion—Prepubertal or Not?

An early experiment using prepubertal male dogs was performed with daily subcutaneous injections of a GnRH agonist over 23 months [10]. The dogs aged 4.6–5.3 months were examined thoroughly before the beginning of treatment and blood was taken for measurement of progesterone, 17-OHprogesterone, DHEA, Δ5-diol, androstenedione, testosterone, DHT, 3α-diol, 3ß-diol, and E2. They were found prepubertal at the beginning of the experiment and no flare-up was observed during treatment. This study shows that the age should not be the only factor to be considered when postponement of puberty is planned. A thorough examination should be mandatory.

The first deslorelin implants were applied by Sirivaidyapong et al. [9]. This group used 4.7 and 9.4 mg deslorelin implants in 4-month-old prepubertal dogs. Unfortunately, authors did not describe how puberty was excluded; the male dogs were just observed by the owners for sexual behavior and only dogs not showing this behavior received the implants.

In another study [23], male puppies received two 9.4 mg deslorelin implants or placebos subcutaneously within the first 24 h after birth. The puppies were examined once weekly for signs of puberty; no flare-up occurred.

### 2.13. Duration of Postponement

In the study by Lacoste et al. [10], the secretion of sexual hormones was effectively suppressed during the treatment with a GnRH agonist over 23 months. After the end of treatment, three male dogs were euthanized and the testicles, prostate, and pituitary glands examined histologically. The testicles showed retarded development of seminiferous tubules and absence of germ cells. The other dogs were observed until puberty, which occurred one month after cessation of treatment when all measured hormones had reached values typical for adults. The testicles of these dogs showed complete regeneration with normal spermatogenesis at 14 months after cessation of treatment. The pituitary–testicular axis therefore showed normal function and fertility was proven by successful matings of two treated males [10]. This injection therapy was therefore considered successful for suppression of puberty; however, development of subcutaneous implants clearly improved the situation for the dogs.

In another study [9], duration of postponement was 34 months with the 4.7 mg implants and >34 months with the 9.4 mg implants. The first semen collection was done at the age of eight months and the maximum diameter of the testicles was measured with a caliper. Both measures were repeated at 12, 15, 18, 24, 30, 32, 34, and 36 months of age. The beagles and mongrels were observed until the testicles had reached a diameter of >2 cm and until occurrence of sexual behavior; unfortunately, the body weight is not given. In three out of four dogs with 4.7 mg deslorelin, sexual behavior was observed at 34 weeks of age and a normal ejaculate collected in two out of these; one dog just showed mounting behavior. On the other hand, all dogs with 9.4 mg (4/4) showed mounting behavior only at this time point; two months later, sexual behavior was still suppressed in two dogs and no semen was collectable in any dogs. Therefore, the suppression was successful and even longer in dogs that received 9.4 mg deslorelin, since the controls showed sexual behavior with penile erection, libido, mounting, or mating behaviors at the age of 8–12 months and produced normal ejaculates. However, the dose-dependent effect still has to be proven with more patients.

In male neonatal puppies, treated subcutaneously with 18.8 mg deslorelin at the age of one day, semen collection started at the age of four months [23]. Sexual behavior was observed by the owners for 1 h twice daily. Puberty was defined as the appearance of both the typical sexual behavior [15] in addition to spermatozoa at semen collection [15,16]. Measurement of testosterone started at the age of one week. The implants suppressed puberty until the age of on average 17 months. In comparison, the controls reached puberty in half the time period. However, since the standard deviation of the age at puberty in the experimental group equals the age at puberty of the controls, the authors of this review suggest that in some animals, the implant failed.

### 2.14. Body Development

Similar to the findings in female dogs, postponement of puberty with GnRH agonists did not cause developmental problems in male dogs. In all studies, height at withers and body weight were similar in experimental and control dogs [9,10,23]. The reason for this phenomenon can probably be explained with the same local effects as mentioned in the chapter about female dogs.

### 2.15. Miscellaneous Effects

Side effects were not observed. A flare-up did not occur in any dog. Unfortunately, epiphyseal closure was not controlled in any study [9,10,23].

Furthermore, as in the section about miscellaneous effects in female dogs, we wish to emphasize that many effects of the GnRH agonists in prepubertal male dogs should be better investigated. As in prepubertal female dogs, the LH and FSH concentrations stay basal during postponement of puberty with GnRH agonists, which might prevent some diseases; however, this still has to be proven [14,31].

### 2.16. Effects on Behavior in Male Dogs

Despite the widespread view that surgical or medical sterilization at an early age eliminates problematic behavior in both male and female dogs [39,40], recent studies show that this view may not be valid for some behavioral problems, namely those not caused by testosterone. There is even some concern that prepubertal sterilization may increase the likelihood of behavioral disorders [41,42,43]. However, almost all of these findings have been compiled from personal observations of animal owners. The individual variability of such personal data, which does not include species-specific standard behavioral tests, makes the evaluation difficult.

In adult male dogs, deslorelin has been reported to provide a significant improvement in behavior in 24 out of 26 dogs with hypersexuality, whereas in terms of hypersexuality and aggressive behavior problems, 10 out of 19 dogs improved [44]. There are limited observations about behavioral changes in deslorelin-implanted, prepubertal male dogs. After the application of different doses of deslorelin (9.4 mg and 4.7 mg) in prepubertal male dogs aged 4 months (n = 8), sexual behavior was not reported until 34 months of age [9]. Therefore, prepubertal GnRH agonist implants can be considered a useful alternative to surgical castration, suppressing any testosterone-mediated behaviors in male dogs [45]. At the end of the duration period, owners can observe their dogs and decide whether castration is an option. However, surgical or medical sterilization of dogs with sociopathic disorders and displaying intraspecific (dog-to-dog) and/or interspecific (dog-to-another) aggression/anxiousness is an individual-specific issue that should be carefully evaluated. A thorough individual analysis of the cause of aggression is obligatory before any of these measures is taken.

### 2.17. To Summarize the Current Knowledge on Deslorelin Implants in Male Dogs

The age at implant insertion should be between 3 and 4 months for small- and medium-sized dogs.

Puberty should be excluded by clinical examination inclusive of testicular palpation, testosterone measurement, and lack of sexual behavior.

The concentration of deslorelin is probably related to the duration of efficacy, which has to be proven.

It is not necessary to remove the implant.

Body development was normal in all studies.

## 3. Conclusions

Age is an important factor concerning the effect of the GnRH analogues, both in male and female dogs. When prepubertal insertion is planned, dogs should be examined accurately for signs of puberty before the application to avoid a flare-up. Nevertheless, it should be considered that the duration of efficacy and the effects on behavior will be affected by individual and breed-related variables. Undesired effects are mostly caused by a flare-up, when the implant is inserted at the age of 7 months or later. In future studies, homogenous groups should be used, especially concerning the confirmed prepubertal stage and the body weight; furthermore, detection of flare-up symptoms should include vaginal cytology and hormonal measurements. In addition, larger patient groups and long-term observations are desirable and urgently needed to provide evidence concerning the impact of delay of puberty on the development and health of bitches.

## Figures and Tables

**Table 1 animals-12-02267-t001:** Experiments using GnRH agonists for postponement of puberty in female dogs.

Animals (N = Bitches)	Age at Implantation (Months)	Body Weight (kg)/Breed	Initial Examination: Exclusion of Puberty	GnRH Agonist	Signs of Flare up	^1^ Time to Puberty ^2^ Age at Puberty (Months)	Controls	Observation Period (Months)	Removal of Implant after (Months) after Insertion	Body Development	Authors
1	4	Beagles		Nafarelin (s.c. osmotic pumps, placed monthly)	-	^1^ 12	^2^ 11	12	-		[17,21]
9	4.6–5.3	1.5–3	Clinical P4, 17-OHprogesterone, DHEA, Δ5-diol, ASD, T, DHT, 3α-diol, 3ß-diol and E2	[D-Trp6, des-Gly-NH_2_^10^] GnRH ethylamide, 100 µg in 0.9 NaCI-1% gelatin Daily ge2.5. latin injections, for 23 months	-	^1^ 4 and 6 (n = 2, all others within the first year) after cessation of treatment		23	-	Normal body weight	[10]
4	4.6–5.3	1.5–3	Clinical P4, 17-OHprogesterone, DHEA, Δ5-diol, ASD, T, DHT, 3α-diol, 3ß-diol, and E2	Controls: same procedure. Vehicle alone	-		-	23	-	Normal body weight	[10]
10	4.88 ± 0.32	Beagles	Vulvar swelling Vaginal bleeding	18.5 mg azagly-nafarelin	-	^2^ 25.5 (18–31)		24	12	Height at withers, body weight normal	[7]
10	4.88 ± 0.32	Beagles	Vulvar swelling Vaginal bleeding	Placebo	-		^2^ 11.9 (8–16)			Height at withers, body weight normal	[7]
6	4	-	Not described	9.4 mg deslorelin	-	^1^ >9		9	-	Not described	[8]
6	7	-	Not described	9.4 mg deslorelin	6/6: Vulvar swelling, vaginal discharge, 4/6 SCI↑, P4 ↑ ( > 2 ng/mL)	^1^ 1–2 weeks		9	-	Not described	[8]
6	4	-	Not described	Placebo	-		^1^ 5/6: 3–7	9	-	Not described	[8]
15	2.8–5.3	3–40	Not described	4.7 mg deslorelin	-	^1^ 13–24		16–25	-	Growth, genital organs normal	[22]
11	2.8–5.3	3–40	Not described	9.4 mg deslorelin	-	^1^ >8–15		8–15	-	Growth, genital organs normal	[22]
8	4.5 + 9 + 13.5	7.6 ± 0.9	Vulvar swelling Vaginal bleeding Vaginal cytology Sonography, E2, P4	4.7 mg deslorelin	Cornification of superficial cells, 50–80%, E2 ↑	^1^ >18		18 (then castration)	-	Juvenile vulva, narrow vaginal channel Normal body weight	[18]
8	4.5	7.4 ± 0.8	Vulvar swelling Vaginal bleeding Vaginal cytology	Controls (1) No implant	-		^1^ 1st estrus: 6–13 ^1^ 2nd estrus: 12–17	18 (then castration)	-	Normal external genitalia Normal body weight	[18]
8	4.5	7.7 ± 1.0	Vulvar swelling Vaginal bleeding Vaginal cytology	Controls (2) No implant	-		-	4.5 (castration)	-	Normal body weight	[18]
4	4.3 (4–5.1)	6–15	Vulvar swelling Vaginal bleeding Vaginal cytology Sonography, E2, P4	4.7 mg deslorelin	1/4: E2 ↑ 2/4: SCI ↑ (15–20%) + E2 ↑	^1^ ~18–>25		~2	-	Normal external genitalia Delayed epiphyseal closure	[11,20]
5	4.2 (4–5.1)	6–15	Vulvar swelling Vaginal bleeding Vaginal cytology Sonography, E2, P4	9.4 mg deslorelin	2/5: P4 ↑ >1ng/mL 1/5: E2 ↑ 2/5: SCI ↑ (15–20%) + E2 ↑	^1^ ~ 21–>25 (2/5)		~25	-	Normal external genitalia Delayed epiphyseal closure	[11,20]
4	4.2 (4–5.1)	6–15	Vulvar swelling Vaginal bleeding Vaginal cytology Sonography, E2, P4	Controls 0.9% NaCl	-		^1^ ~13–18	~25	-	Normal external genitalia Epiphyseal closure at puberty	[11,20]
12	≤24 h m + f	Progenitors: Cross-bred 13 ± 1.9 kg	Sexed and weighed E2 (serum) after 1 week	18.8 mg deslorelin (two 9.4 mg implants)	none	^2^ 72 ± 8.7 weeks (~18 months)		>100 weeks	-	Normal withers height& body weight	[23]
12	≤24 h m + f	Progenitors: Cross-bred 13 ± 1.9 kg	Sexed and weighed E2 (serum) after 1 week	Placebo implants	none		^2^ 39.3 ± 1.2 weeks	>100 weeks	-	Normal withers height& body weight	[23]

ASD = Androstenedione, DHEA = dehydroepiandrosterone, DHT = dihydrotestosterone, E2 = 17ß-estradiol, f = female, m = male, P4 = progesterone, T = testosterone, – = not described. ^1^ Time to puberty. ^2^ Age at puberty.

**Table 2 animals-12-02267-t002:** Experiments using GnRH agonists for postponement of puberty in male dogs.

Animals (N = Male Dogs)	Age at Implantation (Months)	Body Weight (kg)/Breed	Initial Examination/ Exclusion of Puberty	GnRH Agonist	Signs of Flare up	^1^ Time to Puberty ^2^ Age at Puberty (Months)	Controls	Observation Period (Months)	Body Development	Authors
6	4.6–5.3	1.5–3	Clinical P4, 17-OHprogesterone, DHEA, Δ5-diol, ASD, T, DHT, 3α-diol, 3ß-diol, and E2	[D-Trp6, des-Gly-NH_2_^10^] GnRH ethylamide, 100 µg in 0.9 NaCI-1% gelatin. Daily s.c. injections, for 23 months	-	^1^ 1 month after cessation of treatment		23	Normal body weight	[10]
4	4.6–5.3	1.5–3	Clinical P4, 17-OHprogesterone, DHEA, Δ5-diol, ASD, T, DHT, 3α-diol, 3ß-diol, and E2	Controls: Vehicle alone	-	-	-	23	Normal body weight	[10]
4	4	Two Beagles, two mongrels	Observation of sexual behavior	4.7 mg deslorelin		34	-	36	No difference between groups	[9]
4	4	Two Beagles, two mongrels		9.4 mg deslorelin		>34				
3	4	Two Beagles, one mongrel		Controls (Placebo)		8–12				
12	≤24 h m + f	Progenitors: Cross-bred 13 ± 1.9 kg	Sexed and weighed, T (serum) starting after 1 week	18.8 mg deslorelin (two 9.4 mg implants)		^2^ 70.7 ± 35.5 weeks (~17 months)			Normal withers height and body weight	[23]
12	≤24 h m + f	Progenitors: Cross-bred 13 ± 1.9 kg	Sexed and weighed T (serum) starting after 1 week	Placebo implants			^2^ 35.2 ± 2.5 weeks		Normal withers height and body weight	[23]

ASD = Androstenedione, DHEA = dehydroepiandrosterone, DHT = dihydrotestosterone, E2 = 17ß-estradiol, f = female, m = male, P4 = Progesterone, T = testosterone, – = not described. ^1^ Time to puberty. ^2^ Age at puberty.

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
