# Peer review of "Prepubertal Use of Long-Term GnRH Agonists in Dogs: Current Knowledge and Recommendations"

_animals, 2022, doi:10.3390/ani12172267_

Round 1
Reviewer 1 Report
Dear authors,
I don't have many comments or suggestions for changes because your review is very well written and very complete.
However I see two important things that are missing :
- paragraph 2.1. Puberty (lines 76 to 92): for me it is absolutely mandatory to add a few sentences about the interest/ or not / of assaying AMH as a marker of puberty (see Walter 2020 and others);
-even more important according to me: before the conclusion, you should write and add a DISCUSSION on what should be done - in detail - to better study in the future the short, medium and long term effects of the prepubertal use of GnRH agonists. This is really missing. It is indeed a pity, after having analyzed so well the scientific literature, the qualities and failures of the published articles and the missing data in this topic, not to propose concretely, in your opinion, what should/could be done. This would be scientifically judicious and very useful, and your review would very much benefit from it. Otherwise your review, which is interesting of course, remains a bit bland and loses its interest. Well this is just my opinion.
Otherwise I have noticed some tiny things:
-line 87: personnally I would be scared to practice vaginoscopy in prepubertal bitches. Do you confirm this ? And if so, could you tell the reader what to expect to see in case of a prepubertal bitch / or a post-pubertal bitch ?
-line 100: it is 5.3 months, not years;
-lines 137-138: I don't understand why you mention the study of Salmeri et al. which concens the effects of surgical gonadectomy. You mix it with other studies using GnRH agonists. It makes your words confusing and ambiguous and the reader is a bit lost. What are you trying to demonstrate by using this reference? In my opinion, either don't mention it, or explain better what you want to enlight, making it clearer that it has nothing to do with GnRH agonists;
-line 263: "the implant must not be removed": this concerns deslorelin SC implants and should be mentionned. Indeed there are other sources of GnRH agonists that you have mentioned (repeated injections, azagly-nafarelin etc...) - line 257: may be should be rephrased: "to summarize the current knowledge about the use of deslorelin SC implants in prebubertal female dogs" (suggestion);
- line 336: "measurement of testicular diameter": I am very doubtful as there are no tables of references of what to expect, knowing the vast diversity of sizes between a chihuahua to a mastiff;
-line 336 ans line 262: it should be discussed somewhere why, in your opinion, the concentration of deslorelin seems to be related to the duration of efficacy in prebubertal male dogs but not in females.
Congratulations and best regards.
Author Response
I see two important things that are missing :
- paragraph 2.1. Puberty (lines 76 to 92): for me it is absolutely mandatory to add a few sentences about the interest/ or not / of assaying AMH as a marker of puberty (see Walter 2020 and others).
Authors: We inserted a respective paragraph:
“Measurement of AMH can be helpful; however, in one study, 50% of female dogs aged 3-6 months had a serum-AMH concentration higher than the reported cut-off value, the other 50% were below this value. At the age of 6 months, 93.9% had a serum-AMH concentration higher than the reported cut-off value [23]. Therefore, the measurement of AMH seems to be more reliable in bitches older than 6 months as a test for intactness than as a proof of prepubertal ovaries. The latter should be confirmed by examination of more prepubertal bitches.”
-even more important according to me: before the conclusion, you should write and add a DISCUSSION on what should be done - in detail - to better study in the future the short, medium and long term effects of the prepubertal use of GnRH agonists. This is really missing. It is indeed a pity, after having analyzed so well the scientific literature, the qualities and failures of the published articles and the missing data in this topic, not to propose concretely, in your opinion, what should/could be done. This would be scientifically judicious and very useful, and your review would very much benefit from it. Otherwise your review, which is interesting of course, remains a bit bland and loses its interest. Well this is just my opinion.
Authors: we agree and some remarks were already made in the previous manuscript. Especially the proper examination of dogs concerning the prepubertal stage was highlighted. Without this measure, future studies concerning the short, medium and long term effects are useless. Respective remarks were made in the overworked manuscript, where appropriate (highlighted in yellow).
Furthermore the suggestions are summarized in the conclusions:
“In future studies, homogenous groups should be used, especially concerning the confirmed prepubertal stage and the body weight; furthermore, detection of flare-up symptoms should include vaginal cytology and hormonal measurements. In addition, larger patient groups and long-term observations are desireable and urgently needed to provide evidence concerning the impact of delay of puberty on the development and health of bitches.”
Otherwise I have noticed some tiny things:
-line 87: personnally I would be scared to practice vaginoscopy in prepubertal bitches. Do you confirm this ? And if so, could you tell the reader what to expect to see in case of a prepubertal bitch / or a post-pubertal bitch ?
Authors: we inserted a respective paragraph:
“Using a tiny, wettened vaginoscope, it is possible to evaluate, whether the mucus membranes are edematous or not and by taking a swab it is possible to do a cytological examination; an increase in the superficial cells will indicate the beginning of cytological estrus [18].”
-line 100: it is 5.3 months, not years;
Authors: this was corrected
-lines 137-138: I don't understand why you mention the study of Salmeri et al. which concens the effects of surgical gonadectomy. You mix it with other studies using GnRH agonists. It makes your words confusing and ambiguous and the reader is a bit lost. What are you trying to demonstrate by using this reference? In my opinion, either don't mention it, or explain better what you want to enlight, making it clearer that it has nothing to do with GnRH agonists;
Authors: We aimed to explain that surgical prepubertal castration completely prevents the effect of sexual steroids on the epiphyseal closure and causes the increase in height at withers. In contrary, when bitches receive the GnRH agonist prepubertally, a local effect of sexual steroids cannot be fully excluded since in some cases, an increase in the serum concentration of E2 was observed during the flare-up period. This may be the reason why bitches had normal hight at withers after chemical delay of puberty. However, we deleted the sentence to make the paragraph better understandable and inserted the following paragraph:
“A local effect of sexual steroids on the growth plate during the flare-up period can not be excluded when GnRH agonists are used for postponement of puberty. This may vary between individuals as the flare-up symptoms are individual, but as a matter of fact, the average height at withers was not increased after the treatment”.
-line 263: "the implant must not be removed": this concerns deslorelin SC implants and should be mentionned. Indeed there are other sources of GnRH agonists that you have mentioned (repeated injections, azagly-nafarelin etc...)
Authors: we agree and already clarified this in the title: “To summarize the current knowledge on deslorelin in female dogs”
- line 257: may be should be rephrased: "to summarize the current knowledge about the use of deslorelin SC implants in prebubertal female dogs" (suggestion);
Authors: as mentioned above, we already clarified this in the title.
- line 336: "measurement of testicular diameter": I am very doubtful as there are no tables of references of what to expect, knowing the vast diversity of sizes between a chihuahua to a mastiff
Authors: we agree and changed the recommendation as follows:
“-puberty should be excluded by clinical examination inclusive testicular palpation, testosterone measurement, lack of sexual behaviour”
-line 336 ans line 262: it should be discussed somewhere why, in your opinion, the concentration of deslorelin seems to be related to the duration of efficacy in prebubertal male dogs but not in females.
Authors:we agree and inserted a paragraph at the end of chapter 2.2.2.:
“It is still unclear, whether in bitches, a dose-efficacy relationship exists, since the duration of efficacy was similar with 4.7 and 9.4 mg deslorelin implants [11,14,18,25,31]”
We in addition changed the wording in the summary of the current knowledge (after chapter 2.2.4.)
- the concentration of deslorelin seems not to be related to the duration of efficacy
Reviewer 2 Report
This was a painstaking and comprehensive review of publications on the topic, their merits and their problems. I suggest defining "flare up", I assume it is the induction of estrus or partial induction of estrus, as seen when GnRH agonists are used for that purpose and why they are stopped when vaginal cytology indicates proestrus. Also, define "long-term" delay of puberty (as long as medication given? Months? Years?) Clarification of the use of agonists to delay puberty without affecting body size or growth plate closure is needed. Line 61 is unclear, how can epiphyseal closure delay not result in an impact on body weight or growth? Line 152 is confusing as bitches were age 4.2 years and clearly already post pubertal but this paper describes the use of deslorelin for delaying puberty? Lines 264 and 339 should be clarified "mostly" normal is vague. What wasn't normal? Line 293 states testicular atrophy but this was in prepubertal dogs whose testes would theoretically not be mature and able to atrophy? Please clarify. Conclusions: please elaborate on why prepubertal insertion performed "too late i.e. post early puberty, is such a problem? With continued administration cyclicity will be shut down anyway?
Author Response
This was a painstaking and comprehensive review of publications on the topic, their merits and their problems. I suggest defining "flare up", I assume it is the induction of estrus or partial induction of estrus, as seen when GnRH agonists are used for that purpose and why they are stopped when vaginal cytology indicates proestrus.
Authors: we agree and inserted a respective paragraph:
“The definition of flare-up is inconsistent; in some studies, it is just an increase in cornification of superficial cells and an increase in serum-estradiol (E2) concentrations [14,18,24], in others vulvar swelling and/or vaginal discharge, sometimes combined with an increase in superficial cell index (SCI) and/or an increase in serum-progesterone concentrations [11,14,24].”
Also, define "long-term" delay of puberty (as long as medication given? Months? Years?)
Authors: a definition is given in the introduction:
“The development of slow release GnRH agonists has a long history, but meanwhile subcutaneous implants are available, supposed to exert a long-term effect over several months, for fertility control of male and female dogs.”
Clarification of the use of agonists to delay puberty without affecting body size or growth plate closure is needed.
Authors: we hypothesize the following:
“A local effect of sexual steroids on the growth plate during the flare-up period can not be excluded when GnRH agonists are used for postponement of puberty. This may vary between individuals as the flare-up symptoms are individual, but as a matter of fact, the average height at withers was not increased after the treatment”.
Line 61 is unclear, how can epiphyseal closure delay not result in an impact on body weight or growth?
Authors: we tried to argue with the increase in E2 in some bitches during the flare-up and the effect on the growth plate – see previous paragraph
Line 152 is confusing as bitches were age 4.2 years and clearly already post pubertal but this paper describes the use of deslorelin for delaying puberty?
Authors: we appologize for the typo; of course we ment 4.2. months
Lines 264 and 339 should be clarified "mostly" normal is vague. What wasn't normal?
Authors: we agree and deleted mostly – and changed the wording:
“body development was normal in all studies, when only one 4.7 or 9.4 mg implant was used”
Line 293 states testicular atrophy but this was in prepubertal dogs whose testes would theoretically not be mature and able to atrophy? Please clarify.
Authors: this is reasonable and we changed the wording to “ retarded development of seminiferous tubules and absence of germ cells”
Conclusions: please elaborate on why prepubertal insertion performed "too late i.e. post early puberty, is such a problem? With continued administration cyclicity will be shut down anyway?
Authors: we refer to the term delay of puberty. If somebody wishes to do this, a flare-up is not desired. We do not think that a flare-up is a problem per se; however, the course of an induced estrus may be abnormal and when the initial aim was to postpone puberty, this can be a problem for the vet.
Reviewer 3 Report
The present review wants to sum up the current knowledge about the use of GnRH agonists in prepubertal phase. The topic is of clinical interest, however the paper is not written in a comprehensive way and some parts are just “touched” without being explained. Other than this, some parts are cited in the abstract and conclusions but it is not treated or cited in the text (i.e. the long-term effects on health profiles). The paper should be revolutionized and some suggestions are present in the comments below.
English language should be reviewed.
In the present reviewer’s opinion, the paper in its present form is not suitable for being published, it has to be thoroughly ameliorated and improved.
Line 20: remove the comma after “dogs”.
Line 38: remove some spaces before the squared brackets.
Line 47: remove the comma after the word “interest”.
Line 57: “However, prepubertal gonadectomy causes delayed closure of growth plates in cats and dogs [16,17], which may predispose the animals among others to orthopedic problems and epiphyseal fractures.”. This sentence is important for the discussion but the reviewer suggests to use it at the beginning of this part, i.e. at line 55. In this way the discussion starts with the state of art of the “standard” surgical approach and then continue with the advantage/disadvantage of the use of deslorelin.
Line 78: “The age at puberty is highly variable, even within breeds and independent on body weight”. Add a reference.
Line 80: keep in mind the difference between puberty and sexual maturity, two different concepts - and indeed later you refer to the single spermatozoa in urines or ejaculate.
Line 83: better to specify because it seems in contrast with the sentence at the line 80.
Line 89: also hormone stimulating tests.
Line 96: “Variable parameters within the study design” change with “the high variability of parameters considered in the different study designs”.
Line 100: are you sure they were implanted at day one? Referring to paper 24.
Line 105: this sentence is too informal, rephrase in a more formal manner, it is for a scientific paper.
Line 124: It seems like the last sentence is not completed, please deepen better.
Section 2.2.1. Age at implant insertion - prepubertal or not? Body development
I suggest to separate the sections, i.e. one dedicated to the pubertal status of the dogs and one about the body development.
I also suggest to better focus the discussion: at first focus on the age at which the implant was inserted, making comparisons among the different works, then focus on the different hormone analyses used, then about the anatomical aspects or clinical examinations done, and so on.
Table 1 is very well done.
Line 188: in what sense it was exceptional?
Line 193: use the proper abbreviation for testosterone, avoid abbreviation for the word “respectively”.
Line 202: this is not clear, as it is described indeed it appears that flare-up is present when there is no ovarian/follicular activity.
Line 206: remove some extra spaces after the full stop (and throughout the rest of the text, there are many).
Line 206-207: please rephrase, the sentence is not clear.
Line 211: ok but the sentence is not well deepened and harmonized with the rest of the paragraph.
Section 2.2.2. Flare-up or not? Duration of postponement
Even in this section I suggest to divide the two main topics, detailing if the flare-up appeared and what was the duration of the postponement in the different studies, even considering the different patients enrolled and the different location of insertion, weight, etc, where possible (if they are indicated in the studies).
Line 234: after “groups” insert the citation.
Section 2.2.4. Miscellaneous effects
Please rephrase and format the paragraph. Insert citations also.
Line 283: remove “at least”.
Section 2.3.1.
The same as suggested for the first part it is suggested for the part referred to males.
Line 303: remove the comma after the word “describe”.
Line 328: please specify if this conclusion is made by the authors of the study or by the authors of the present review.
Section 2.3.2. Miscellaneous effects
Please format and rephrase, adding citations.
Line 356: correct the citations in the brackets. Also, it would be interesting to deepen this part comparing the different findings of the studies cited.
Line 358-370: sexual behaviors and behavioral issues should be discussed separately as they are two completely different topics.
Line 390: “The long-term effects of delay of puberty with deslorelin on joint health, tumor development, the immune system and social behavior remain to be investigated.”. This part is completely missing the main text of the paper and should be thoroughly discussed, as some reports are now existing. Also about the high levels of LH.
Author Response
The present review wants to sum up the current knowledge about the use of GnRH agonists in prepubertal phase. The topic is of clinical interest, however the paper is not written in a comprehensive way and some parts are just “touched” without being explained. Other than this, some parts are cited in the abstract and conclusions but it is not treated or cited in the text (i.e. the long-term effects on health profiles). The paper should be revolutionized and some suggestions are present in the comments below.
English language should be reviewed.
In the present reviewer’s opinion, the paper in its present form is not suitable for being published, it has to be thoroughly ameliorated and improved.
Line 20: remove the comma after “dogs”.
Authors: it was removed
Line 38: remove some spaces before the squared brackets.
Authors: it was removed
Line 47: remove the comma after the word “interest”.
Authors: we removed it
Line 57: “However, prepubertal gonadectomy causes delayed closure of growth plates in cats and dogs [16,17], which may predispose the animals among others to orthopedic problems and epiphyseal fractures.”. This sentence is important for the discussion but the reviewer suggests to use it at the beginning of this part, i.e. at line 55. In this way the discussion starts with the state of art of the “standard” surgical approach and then continue with the advantage/disadvantage of the use of deslorelin.
Authors: we removed this sentence from the introduction
Line 78: “The age at puberty is highly variable, even within breeds and independent on body weight”. Add a reference.
Authors: we added a reference
Line 80: keep in mind the difference between puberty and sexual maturity, two different concepts - and indeed later you refer to the single spermatozoa in urines or ejaculate.
Authors: this was exactly what we wanted to elaborate and underpinned by citing the relevant literature.
Line 83: better to specify because it seems in contrast with the sentence at the line 80.
Authors: We changed the wording as follows and hope this contributes to clarification:
“In males, it is the age at which the first full ejaculate can be collected [18]. However, as reviewed by [19], puberty probably starts earlier, in females, marked by follicular growth, a change in vaginal cytology and the increase in steroid hormones or precursors like dihydroepiandrosterone DHEA [13,20]. In males, it is sometimes marked by beginning of mounting or single spermatozoa in the urine or ejaculate [19].”
Line 89: also hormone stimulating tests.
Authors: we do not agree that stimulating tests are necessary for this puprose
Line 96: “Variable parameters within the study design” change with “the high variability of parameters considered in the different study designs”.
Authors: we changed the sentence accordingly.
Line 100: are you sure they were implanted at day one? Referring to paper 24.
Authors: the implants were in fact implanted at day one.
Line 105: this sentence is too informal, rephrase in a more formal manner, it is for a scientific paper.
Authors: we changed the sentence to
“And some studies were performed without a control group [13,24].”
Line 124: It seems like the last sentence is not completed, please deepen better.
Authors: we changed the sentence:
“This study highlighted the efficacy of long-term applied GnRH agonists; however, later on the long-acting subcutaneous GnRH-agonist implants were introduced and proved to be highly effective in many studies.”
Section 2.2.1. Age at implant insertion - prepubertal or not? Body development
I suggest to separate the sections, i.e. one dedicated to the pubertal status of the dogs and one about the body development.
I also suggest to better focus the discussion: at first focus on the age at which the implant was inserted, making comparisons among the different works, then focus on the different hormone analyses used, then about the anatomical aspects or clinical examinations done, and so on.
Authors: we separated the chapters as suggested
Table 1 is very well done.
Line 188: in what sense it was exceptional?
Authors: in that so young animals were used. We changed the wording to clarify this:
“In the study of Faya et al [23], cross-bred dogs with an average body weight of 13 kg were mated, and the newborn male and female puppies were randomly assigned to experimental and control group at the age of one day.”
Line 193: use the proper abbreviation for testosterone, avoid abbreviation for the word “respectively”.
Authors: we changed this accordingly
Line 202: this is not clear, as it is described indeed it appears that flare-up is present when there is no ovarian/follicular activity.
Authors: we wanted to highlight that in some dogs, there is already an increase in FSH and follicular development starts, even at the age of 4 months. To clarify this, we changed the wording.
“A flare-up with visible effects of estrogens was described in 4/7 studies in female dog, indicating that in some dogs, FSH secretion and follicular development had already started. “
Line 206: remove some extra spaces after the full stop (and throughout the rest of the text, there are many).
Authors: this was done
Line 206-207: please rephrase, the sentence is not clear.
Authors: we rephrased as follows:
“ Bloody vaginal discharge indicating estrus bleeding only occured, when puppies aged 7 month received a subcutaneous GnRH-agonist implant [7]; this can be confirmed by unpublished results (Kaya et al. unpublished).”
Line 211: ok but the sentence is not well deepened and harmonized with the rest of the paragraph.
Authors: we changed the sentence:
“In the study of Faya et al [23], none of the newborn puppies that had received subcutaneous deslorelin implants [23] at the age of one day showed flare-up symptoms.”
Section 2.2.2. Flare-up or not? Duration of postponement
Even in this section I suggest to divide the two main topics, detailing if the flare-up appeared and what was the duration of the postponement in the different studies, even considering the different patients enrolled and the different location of insertion, weight, etc, where possible (if they are indicated in the studies).
Authors: we completely rearranged the chapters, following the suggestion.
Line 234: after “groups” insert the citation.
Authors: the citation was inserted
Section 2.2.4. Miscellaneous effects
Please rephrase and format the paragraph. Insert citations also.
Authors: we rephrased the chapter and added a paragraph about possible, beneficial effects (taken out of the introduction, where it did not fit)
“The consecutive decrease in the blood concentration of FSH and LH is one important difference when the effect of prolonged administration of GnRH agonists is compared to gonadectomy. After gonadectomy, serum-concentrations of FSH and LH stay elevated for years if not decades. The probably negative, local effects of these increased concentrations on different organs and cancer development are currently under investigation (Ettinger et al., 2019; Kutzler, 2020). It is of interest whether the long-term suppression of cyclicity with slow-release GnRH agonists influences the incidence of some of the mentioned diseases, probably related to increased LH receptor expression after gonadectomy (Ettinger et al., 2019). This would be a desirable effect in male and female dogs; however, further studies and observation of treated dogs over years are necessary to provide evidence.”
Line 283: remove “at least”.
Authors: this was done
Section 2.3.1.
The same as suggested for the first part it is suggested for the part referred to males.
Authors: we did as suggested
Line 303: remove the comma after the word “describe”.
Authors: we deleted it
Line 328: please specify if this conclusion is made by the authors of the study or by the authors of the present review.
Authors: we inserted a few words:
“However, since the standard deviation of the age at puberty in the experimental group equals the age at puberty of the controls, the authors of this review suggest that in some animals the implant failed.”
Section 2.3.2. Miscellaneous effects
Please format and rephrase, adding citations.
Authors: we did so
Line 356: correct the citations in the brackets. Also, it would be interesting to deepen this part comparing the different findings of the studies cited.
Authors: We do not agree and think that this is not the aim of this review, since this referres to post-pubertal dogs.
Line 358-370: sexual behaviors and behavioral issues should be discussed separately as they are two completely different topics.
Authors: the conclusions were overworked
Line 390: “The long-term effects of delay of puberty with deslorelin on joint health, tumor development, the immune system and social behavior remain to be investigated.”. This part is completely missing the main text of the paper and should be thoroughly discussed, as some reports are now existing. Also about the high levels of LH.
Authors: as mentioned, the conclusions were changed and these issues discussed in other chapters.
Reviewer 4 Report
Authors created a review of the literature regarding the use of GnRH agonists as an effective methodology to delay puberty in dogs. The review critically examines relevant experiments to draw summarized conclusions on the efficacy of the data and potential clinical recommendations. The information is relevant to the clinical community and may assist guiding clinicians in the potential use of GnRH agonists for clients. Despite the relevance the review the manuscript has major grammatical issues including the usage of incorrect words, sentence structure, and length of sentences. Furthermore, section 2.2.4. is significantly deficient in descriptive information regarding tissues, steroidogenic parameters, and pubertal parameters that were measured in cited studies. Below are some of the specific suggestions, corrections, and suggested revisions.
1. Line 110: replace the word ‘falsified’ with ‘confounded’ or ‘invalidates’ the results. Using the term ‘falsified’ implies that results were deliberately fabricated rather than meaningless.
2. Table 1 is disorganized:
a. Column titles are disorganized. Should center the titles and abbreviate months to mos. First column title should have the ‘N’ in parentheses. Try to avoid breaking up words in the titles.
b. In the authors column, authors for previous publications need to be represented numerically as written in text, not by last name.
c. Some column text is centered and some column text is justified to the left which is inconsistent and messy.
d. Numeric superscripts for data is not defined. A sentence defining the superscripts beneath the table needs to be added.
3. Author references in text are not all accurately written: number is used within text to state an author and respective study, which is not correct. For example, instead of ‘In the study of [13]……’ write ‘In the study conducted by Rubion et al., [13]….’. Similar editing needed for the many sentences. Below is an example of some sentences in the text. A thorough review by authors is required to identify all of the sentences where ref are cited as such:
i. line 81 reference [21] use an author name.
ii. line 101 reference [26] use an author name.
iii. line 139 reference [13] use an author name.
iv. line 142 reference [15] should be an author name OR move the number to the end of the sentence.
v. line 159 reference [13] use an author name
VI. many more instances of incorrect citation in the remaining text.
4. Sentence 188 and 189 needs to be combing to create one sentence. Example: ‘An exceptional study was conducted by Faya et al [24], where newborn cross-bred male and female pups, weighing an average 13kg, were randomly assigned to treatment or control groups’.
5. Line 190: Replace the term ‘experimentals’ with ‘treatment group’. The study is an experiment and the control groups are part of the experiment because it is used to compare the effect of treatment, i.e. the implant.
6. Line 191: was the implant subcutaneous?
7. Line 190-194: Sentence too long. Furthermore, the end of the sentence doesn’t make sense. Were the acronyms ‘E2’ or ‘Tst’ ‘resp’ previously defined?
8. Line 195: Sentence does not make sense. Add ‘in the treatment group (age)’ after the word ‘double’. Add the ages for the treatment group and control groups in parenthesis immediately after the groups, i.e., treatment group (8-10 mos), control groups (4-5 mos).
9. Line 197: replace the word ‘puppy’ with ‘puppies’. Replace experimental with treatment.
10. Line 199: add the word ‘the’ before the word vulva.
11. Line 200: replace the word ‘channel’ with ‘canal’.
12. Line 202: Define a flare-up
13. Line 203 delete ‘the’ before FSH
14. Line 207: implanted with what? Replace word ‘own’ with ‘unpublished’
15. Line 209: delete the word ‘to’.
16. Line 210: replace word ‘on’ with ‘of’
17. Line 211: use author name to help the reader follow the author, i.e., ‘none of the newborn puppies reported by Faya et al. [24] that received…….’
18. Line 213-217: Sentence too long and confusing as written. Please revise.
19. Line 219: replace ‘further’ with ‘other’.
20. Line 227: insert ‘was’ before the word observed.
21. Section 2.2.3: none of the proteins, enzymes, or receptors or tissues analyzed are adequately described regarding steroidogenesis and maturation of the hypothalamic-pituitary-reproductive tissue axis. This section needs a lot more information included to adequately review the influence of treatment methodologies on the process of steroidogenesis and regulators of pubertal development.
22. Section 2.2.4 and 2.3.2: Formatting of the paragraph is problematic. Incomplete sentences, inappropriate paragraphs, and offline indentation.
23. Table 2: See Table 1 comments above.
24. Line 286: dog or dogs?
25. Line 290: revise, ‘were clearly prepubertal’. At the beginning of what? Initiation of treatment administration?
26. Line 291: What is an endocrinium?
27: Line 313: Reword beginning to ‘Therefore, the suppression’.
28. Line 317: Revise sentence and delete ‘in this manuscript’.
29. Line 384-385: Sentence is unclear. Both age and independent behavioral changes in both males and females influence the outcome of GnRH analogues??
30. Line 388: delete ‘as well’.
Author Response
Authors created a review of the literature regarding the use of GnRH agonists as an effective methodology to delay puberty in dogs. The review critically examines relevant experiments to draw summarized conclusions on the efficacy of the data and potential clinical recommendations. The information is relevant to the clinical community and may assist guiding clinicians in the potential use of GnRH agonists for clients. Despite the relevance the review the manuscript has major grammatical issues including the usage of incorrect words, sentence structure, and length of sentences. Furthermore, section 2.2.4. is significantly deficient in descriptive information regarding tissues, steroidogenic parameters, and pubertal parameters that were measured in cited studies. Below are some of the specific suggestions, corrections, and suggested revisions.
- Line 110: replace the word ‘falsified’ with ‘confounded’ or ‘invalidates’ the results. Using the term ‘falsified’ implies that results were deliberately fabricated rather than meaningless.
Authors: the word was changed
- Table 1 is disorganized:
- Column titles are disorganized. Should center the titles and abbreviate months to mos. First column title should have the ‘N’ in parentheses. Try to avoid breaking up words in the titles.
- In the authors column, authors for previous publications need to be represented numerically as written in text, not by last name.
- Some column text is centered and some column text is justified to the left which is inconsistent and messy.
- Numeric superscripts for data is not defined. A sentence defining the superscripts beneath the table needs to be added.
Authors: the table was overworked accordingly
- Author references in text are not all accurately written: number is used within text to state an author and respective study, which is not correct. For example, instead of ‘In the study of [13]……’ write ‘In the study conducted by Rubion et al., [13]….’. Similar editing needed for the many sentences. Below is an example of some sentences in the text. A thorough review by authors is required to identify all of the sentences where ref are cited as such:
- line 81 reference [21] use an author name.
Authors: this was changed
- line 101 reference [26] use an author name.
Authors: this was changed
iii. line 139 reference [13] use an author name.
Authors: This paragraph was changed
- line 142 reference [15] should be an author name OR move the number to the end of the sentence.
Authors: This paragraph was changed
- line 159 reference [13] use an author name
Authors: This paragraph was changed
- many more instances of incorrect citation in the remaining text.
Authors: The citations are correct; however, we named the authors before the numbers, as suggested, throughout the manuscript
- Sentence 188 and 189 needs to be combing to create one sentence. Example: ‘An exceptional study was conducted by Faya et al [24], where newborn cross-bred male and female pups, weighing an average 13kg, were randomly assigned to treatment or control groups’.
Authors: this paragraph was completely changed
- Line 190: Replace the term ‘experimentals’ with ‘treatment group’. The study is an experiment and the control groups are part of the experiment because it is used to compare the effect of treatment, i.e. the implant.
Authors: this was changed
- Line 191: was the implant subcutaneous?
Authors: yes, this was explained in the overworked manuscript
- Line 190-194: Sentence too long. Furthermore, the end of the sentence doesn’t make sense. Were the acronyms ‘E2’ or ‘Tst’ ‘resp’ previously defined?
Authors: this paragraph was changed:
“The treatment group received two 9.4 mg implants subcutaneously in the left and right shoulder blade region. They were examined once weekly for body development (height at withers, body weight, scrotum circumference) and signs of estrus, and blood was taken for estradiol (E2) and testosterone (T) measurement. In addition, the owners looked for sexual behaviour”
- Line 195: Sentence does not make sense. Add ‘in the treatment group (age)’ after the word ‘double’. Add the ages for the treatment group and control groups in parenthesis immediately after the groups, i.e., treatment group (8-10 mos), control groups (4-5 mos).
Authors: this was changed accordingly
- Line 197: replace the word ‘puppy’ with ‘puppies’. Replace experimental with treatment.
Authors: this was changed accordingly
- Line 199: add the word ‘the’ before the word vulva.
Authors: this was changed accordingly
- Line 200: replace the word ‘channel’ with ‘canal’.
Authors: this was changed accordingly
- Line 202: Define a flare-up
This was inserted in the overworked manuscript, at the beginning of the postponement of pubert in female dogs chapter:
“The definition of flare-up is inconsistent; in some studies, it is just an increase in cornification of superficial cells and an increase in serum-estradiol (E2) concentrations (Kaya et al., 2015; Kaya et al., 2013; Marino et al., 2014), in others vulvar swelling and/or vaginal discharge, sometimes combined with an increase in superficial cell index (SCI) and/or an increase in serum-progesterone concentrations (Kaya et al., 2015; Kaya et al., 2013; Trigg et al., 2006).”
- Line 203 delete ‘the’ before FSH
Authors: this was changed accordingly
- Line 207: implanted with what? Replace word ‘own’ with ‘unpublished’
Authors: this was changed accordingly
- Line 209: delete the word ‘to’.
Authors: it was deleted
- Line 210: replace word ‘on’ with ‘of’
Authors: this was changed accordingly
- Line 211: use author name to help the reader follow the author, i.e., ‘none of the newborn puppies reported by Faya et al. [24] that received…….’
Authors: this was changed accordingly
- Line 213-217: Sentence too long and confusing as written. Please revise.
Authors: this paragraph was completely changed
- Line 219: replace ‘further’ with ‘other’.
Authors: this was changed accordingly
- Line 227: insert ‘was’ before the word observed.
Authors: the word was inserted
- Section 2.2.3: none of the proteins, enzymes, or receptors or tissues analyzed are adequately described regarding steroidogenesis and maturation of the hypothalamic-pituitary-reproductive tissue axis. This section needs a lot more information included to adequately review the influence of treatment methodologies on the process of steroidogenesis and regulators of pubertal development.
Authors: we discussed the function of the investigated parameters in the new manuscript and hopefully increased the information:
“The local impact of deslorelin on ovarian suppression and resumption of ovarian activity was examined after the end of suppression of cyclicity (Kaya et al., 2017); investigated were enzymes and hormones involved in the corpus luteum (CL) establishment and function. Finally, no long-term effect on luteal function after the first estrus was observed. Expression of enzymes involved in steroidogenic pathways in the ovaries like steroidogenic acute regulatory protein (STAR) was similar in both groups, even though expression of another enzyme involved in these pathways, namely 3β-hydroxysteroid dehydrogenase (3ßHSD), was significantly increased after deslorelin treatment. However, the progesterone concentration during metestrus following the first estrus after suppression was equal in both groups (Kaya et al., 2017). Similarly, expression of enzymes involved in prostaglandin synthesis pathway like cyclooxygenase 2 (COX-2 and PTGS), did not differ significantly between groups; the prostaglandins are significantly involved in the establishment of CL function in dogs (Kowalewski et al., 2013). Other factors like kisspeptin receptor (kiss1-R) and GnRH-R were not significantly changed in the first cycle following suppression; similarly prolactin receptor (PRLR), which is up-regulated during early- and mid-luteal phase to enable the luteotropic action of prolactin. Progesterone receptor (PGR), estrogen receptor alpha (ERa), and estrogen receptor beta (Erß) were not different between groups either.”
“In a follow up study, the same dogs as in the study of (Kaya et al., 2017) were used; the uterine tissue was examined for expression of GnRH receptor (R), Kisspeptin (KP)10, Kisspeptin receptor (GPR54), androgen receptor (AR), estrogen receptor (ER) α,β, and progesterone receptor (PR) by means of immunohistochemistry and PCR. The treatment obviously had no significant impact on the expression of these receptors important for sexual hormone effectiveness in cycling dogs.”
- Section 2.2.4 and 2.3.2: Formatting of the paragraph is problematic. Incomplete sentences, inappropriate paragraphs, and offline indentation.
Authors: This paragraph was correctly numbered and overworked
- Table 2: See Table 1 comments above.
Authors: the table was overworked accordingly
- Line 286: dog or dogs?
Authors: the word was changed to dogs
- Line 290: revise, ‘were clearly prepubertal’. At the beginning of what? Initiation of treatment administration?
Authors: the paragraph was changed
- Line 291: What is an endocrinium?
Authors: the paragraph was changed
“The dogs aged 4.6-5.3 years were examined thoroughly before the beginning of treatment and blood was taken for measurement of progesterone, 17-OHprogesterone, DHEA, D5-diol, androstenedione, testosterone, DHT, 3a-diol, 3ß-diol and E2. They were found prepubertal at the beginning of the experiment and no flare-up was observed during treatment. This study shows that the age should not be the only factor to be considered when postponement of puberty is planned. A thorough examination should be mandatory.”
27: Line 313: Reword beginning to ‘Therefore, the suppression’.
Authors: this was done
- Line 317: Revise sentence and delete ‘in this manuscript’.
Authors: this was done
- Line 384-385: Sentence is unclear. Both age and independent behavioral changes in both males and females influence the outcome of GnRH analogues??
Authors: we deleted “and including behavioral changes”
- Line 388: delete ‘as well’.
Authors: this was done
Round 2
Reviewer 3 Report
Manuscript has been substantially improved, not all the suggestions were accomplished but in total the manuscript is suitable for publication.
Author Response
We overworked the manuscript according to the suggestion and hope that the english language improved
Reviewer 4 Report
It is appreciated that the authors implemented the numerous reviewer suggestions. There are only a few minor edits required.
1. Line 198: replace ‘experimental’ with ‘treatment’
2. Line 271: Clarify the tissue of the ‘Other factors’ expressed, i.e., ovarian tissue? CL? Include reference for that data.
3. Table 1: Months is inconsistent, ie. mos vs months. Unclear about the title of one column ‘Implant removed after (months)’. Appears incomplete. Should it be ‘duration of implant (months)’.
4. Table 2:
a. Data is misaligned in the ‘Time to puberty’, ‘Controls’, Observation period, and ref column.
b. Months is inconsistent, ie., mos vs months
Author Response
Prepubertal use of long-term GnRH agonists in dogs: current knowledge and recommendations
It is appreciated that the authors implemented the numerous reviewer suggestions. There are only a few minor edits required.
- Line 198: replace ‘experimental’ with ‘treatment’
Authors: this was done
- Line 271: Clarify the tissue of the ‘Other factors’ expressed, i.e., ovarian tissue? CL? Include reference for that data.
Authors: this referres to the study of Kaya et al 2018. We changed the sentence starting in line 265 in the new manuscript:
“Kaya et al [13] investigated enzymes and hormones involved in the corpus luteum (CL) establishment and function. Authors observed no long-term effect on luteal function after the first estrus.”…
And in line 275:
“In the study of Kaya et al [13], kisspeptin receptor (kiss1-R) and GnRH-R were not significantly changed in the first cycle following suppression; similarly prolactin receptor (PRLR),…”
- Table 1: Months is inconsistent, ie. mos vs months. Unclear about the title of one column ‘Implant removed after (months)’. Appears incomplete. Should it be ‘duration of implant (months)’.
Authors: we changed the abbreviation and changed the title of the column:
“Removal of implant after (mos) after insertion”
- Table 2:
- Data is misaligned in the ‘Time to puberty’, ‘Controls’, Observation period, and ref column.
- Months is inconsistent, ie., mos vs months
Authors: this was changed